# Can Molecularly Engineered Plant Galls Help to Ease the Problem of World Food Shortage (and Our Dependence on Pollinating Insects)?

**DOI:** 10.3390/foods11244014

**Published:** 2022-12-12

**Authors:** Victor Benno Meyer-Rochow

**Affiliations:** Department of Ecology and Genetics, University of Oulu, SF-90140 Oulu, Finland; meyrow@gmail.com; Tel.: +358-413-183648

**Keywords:** crop pollinators, pollinator losses, entomophagy, alternative foods, plant galls, bio-engineered nutrition

## Abstract

The world faces numerous problems and two of them are global food shortages and the dwindling number of pollinating insects. Plant products that do not arise from pollination are plant galls, which as in the case of oak apples, can resemble fruits and be the size of a cherry. It is suggested that once research has understood how chemical signals from gall-inducing insects program a plant to produce a gall, it should be possible to mimic and to improve nature and “bioengineer” designer galls of different sizes, colorations and specific contents to serve as food or a source of medicinally useful compounds. To achieve this objective, the genes involved in the formation of the galls need to be identified by RNA-sequencing and confirmed by gene expression analyses and gene slicing. Ultimately the relevant genes need to be transferred to naïve plants, possibly with the aid of plasmids or viruses as practiced in crop productivity increases. There is then even the prospect of engineered plant galls to be produced by plant tissue culture via genetic manipulation without the involvement of insects altogether.

## 1. Introduction: The Problem

Although populations have been decreasing in the majority of the EU-countries [1,2] and in several East Asian countries as well [3], the global population is still rising and expected to reach 10.4 × 10^9^ people by the end of the 21st century [4]. What has been of concern is whether there will be sufficient food for this increase in population and already in 1975 Meyer-Rochow [5] addressed this problem. He wrote in the Australian and New Zealand Association for the Advancement of Science journal *Search* an article with the title “Can Insects Help to Ease the Problem of World Food Shortage?”. In that article he recommended that organizations, such as the FAO and WHO back his idea. Despite the skepticism and even ridicule that this suggestion initially received (Figure 1) [6], it eventually became accepted and over the last 20 years has seen an extraordinary jump in public support—last but not least because of the backing from an FAO publication by Van Huis et al. [7].

Now, nearly 50 years after that pivotal earlier publication on edible insects appeared in Search, I should like to follow up that idea with a suggestion to use molecularly engineered plant galls as a novel food item that does not require the services of pollinating insects and that can be used as a resource from which to extract therapeutically valuable chemicals.

Why should it be important to produce crops that do not require an input from pollinating insects? For a multitude of reasons: in the anthropocene pollinating insect numbers are on the decline worldwide [8,9]. It was noticed already in 2013 that losses of winter bee hives had gone up significantly in both Europe and the USA, and honey harvests in France, for example, had dropped to one third in 2017 of what they used to be in the 1990s [10]. An estimated 87.5% (approximately 300,000 species) of the world’s flowering plants are pollinated by insects [10], and it has been predicted by Giannini et al. in 2017 [11], that disruptions of the pollinating process could lead to crop losses in nearly 90% of the regions of Brazil that were analysed in that study.

Biotic pollination is not only efficient, it actually improves fruit and seed quantity of tropical crops by 70% and crops cultivated in Europe by 85% [12] and although, for example, rice and wheat are wind pollinated, the global economic value of insect pollination was already 153 billion Euros in 2005 [13]. Production losses that can be attributed directly to the absence of flower visitors are of the order of 5% for the developed and 8% for the developing world superimposed on global agricultural production increases of 140% between 1961 and 2006 [12]. This may not sound too bad, but pollinator dependency and the deficits for the developed and the developing regions between 1961 and 2006 rose by 50% and 62%, respectively, and that is quite alarming [12].

There are, of course, other factors that can affect beekeeping and bee populations, as in the case of Romania during the transition period after 1989 to a market economy, when beekeeping experienced a severe decline in that country. For reasons explained in detail by Popescu et al. in 2019 [14], it was largely because of the long history of beekeeping in Romania and the experience in economic, social and environmental contexts that apicultural practices recovered in that country. However, even that country is not immune to the global concerns of declining pollinator populations.

Pollination, of course, in contrast to vegetative reproduction, involves the transfer of pollen, i.e., highly reduced microgametophytes, from the male flower (in case of a dioecious species with separate male and female plants) or the male parts known as the anthers of a bisexual plant, to a flower’s female component, the stigma. What follows, is the fertilisation of the flower’s ovules by the gametes from the pollen grain and the subsequent growth of a fruit with its seed or seeds. Self-fertilisation occurs when pollen from within the same flower or from different flowers but of the same plant get transferred to the female. This can lead to inbreeding and ultimately to a reduced resistance of the plant to diseases and environmental stressors. Therefore it is to be avoided. Cross-pollination in which pollen stems from different plant individuals (or at least different flowers) involves to a very large extent pollinating insects and the latter depend for their survival on a host of different plant species and an intact environment. Bees are the most important pollinators of our fruits, vegetables, nuts, coffee, etc. and their decline together with that of the other pollinators, dealt with in numerous publications [9,10,11,12,13], is therefore of considerable concerns.

Strategies to combat this decline in pollinators exist [15,16], and it has been suggested that the widespread and often uncontrolled use of pesticides can be directly linked to increased insect deaths and loss of insect biodiversity. The increase in transgenic crops to repel insect pests has also been mentioned as a bee killer, because genetically altered organisms are potentially hazardous to bees. Undoubtedly some of the major reasons why we lose pollinators are deforestation, increase of pasture land for cattle and sheep and habitat loss, the latter usually a consequence of urbanisation. Air pollution, global warming and climate change are additional factors that affect insects generally and pollinators in particular. Honey bees, moreover, suffer from attacks by the *Varroa* mite, microsporidial parasites and a variety of diseases. Although creating pollinator-friendly habitats, improving nest sites, curbing the use of insecticides and other strategies have been suggested as counter-measures to stop the loss of pollinating insect species, it is doubtful whether all these measures are feasible and can be implemented. Since it is anybody’s guess whether the suggested measures will actually reverse the trend and how long it might take to build up pollinator populations again, I have started to focus my attention on plant products that often resemble fruits, as noticed already by Darwin see e.g., [17] but which are structures that do not arise from pollination: plant galls (Figure 2).

## 2. Background Information

Plant galls show up as modifications of the normal development of certain plant parts [18,19,20]. They represent abnormal outgrowths that can occur on a plant’s leaves, on stems, flowers and even roots, but apparently do not significantly harm the plant. Structurally plant galls can be very diverse in texture, shape and coloration and may resemble spiky or spikeless pouches, nodules, warts, little round apples or tiny, straight and sometimes twisted chimneys. The production of galls can be a response of the plant to stimuli that stem from viruses, bacteria or fungi, but more importantly it can also be triggered by chemicals released by ovipositing gall insects, the physical damage to the plant that the latter sustains during oviposition by an insect, or salivary and other secretions released by the feeding stages of the developing insects [21,22].

Amongst the insects, major gall inducers are known from the orders of Diptera (families Cecidomyiidae, Tephritidae and Agromyzidae), Hymenoptera (Cynipidae, Chalcidoidae, Tenthredinidae), Coleoptera (Curculionidae, Buprestidae, Cerambycidae), Hemiptera (Apiomorphinae, Aphididae. Coccidae, Psyllidae), Lepidoptera (e.g., *Gnorimoschema gallae solidaginis*) and some Thysanoptera. Other gall-inducing invertebrates are certain mites and nematodes, the latter, in case of *Fergusobia* spp., in a symbiotic relationship with larvae of the fly genus *Fergusonina*. Plant galls may be completely closed, requiring the occupant to make a hole into the gall to get out, or they may have an opening to the outside as with the galls of many aphids and eriophyiid mites [22,23,24].

What all these gall-inducers, especially those of insect origin, have in common is that the gall provides food and shelter for the developing insect stages. A certain advantage for the plant of having the insect restricted to the gall is that the insect cannot ‘move around’ and spread to different parts of the plant: it is trapped. On the other hand, the plant invests energy and resources to produce the gall, while the plant cells’ meristematic tissue proliferates and acts as a powerful “physiologic sink”. This was shown using ^14^C-labelling by Inbar et al. [25], who studied galls formed on *Pistacia palaestina* by aphids of the family Pemphigidae. Starch, sugars and many other chemicals are being diverted from the phloem of the surrounding plant parts to the benefit of the developing insect [26]. Defensive low molecular weight phenolic substances present in the leaves of, for example, willows such as *Salix* spp. [27], are often substantially less prominent in galls, while condensed tannic acid levels, on the other hand, are generally elevated in the gall’s interior and especially its cortex.

It is universally accepted that the formation of a gall is the result of chemicals from the gall inducer, which in the case of the embryo can be hormones that include auxins, cytokinins, indole-3-acetic acid, etc., or are signals from the ovipositing insect and the developing brood that are encoded by transcriptionally co-regulated genes [28]. This means that the gall insect is the ‘driver’ and that the host, i.e., the plant, responds [29] in ways that are not too different from the blastema formation in, for example, the regenerating tail of lizards [30] and the abnormal growths in amphibians [31] as well as the tumours found in vertebrates, generally. Feitelson et al. [32] could show for the latter that a number of natural compounds existed (e.g., curcumin, resveratrol, indole-3-carbinol, brassinin, sulforaphane, epigallocatechin-3-gallate, genistein, ellagitannins, lycopene and quercetin) that would be able to inhibit one or more of the pathways that were known to contribute to the cell proliferation, which can be regarded as the basis for the usually rapid growth of the tumour.

A complete and exhaustive explanation of the mechanism that is involved has yet to be forthcoming [17,32], but it seems certain that with regard to plant galls, it is the gall inducer that manipulates its host to an extent that galls made by the same host varied in nutrients and appearance when different gall-inducing species of insects were involved [33]. Plant galls are therefore the products of a natural kind of genetic engineering, which according to Marx [34], postulated already in 1979, may involve “the transfer of DNA”, although to date such a transfer has not been confirmed and indeed it would be difficult to imagine a mechanism by which this could be achieved.

Numerous studies support the notion that galls are extended phenotypes of the gallers and that the latter rather than the plants determine gall morphology [35,36,37] and Nyman and Julkunen-Tiitto [27] had shown beyond doubt that not only the morphology, i.e., the physical appearance, of the induced galls can be regarded as “an extended phenotype of the galler”, but that the gallers (in their case sawflies that manipulated their willow hosts to form galls on the leaves) can also control the chemical characteristics of the galls. As a result, large volumes of dedicated plant tissue were produced, but the latter contained smaller amounts of phenolic compounds such as flavones, flavonols, salicylates, and cinnamic acid derivatives than the surrounding leaf tissue.

The rapid proliferation of undifferentiated cells that leads to the expansion of the gall, bears similarities to plant tumours [38] and evidence, presented by Schultz et al. [17] suggests that the “hijacking of the underlying genetic machinery in the host plant” by the galler has something to do with the reproductive gene ontology categories as the latter “were significantly enriched in developing galls”. Furthermore, genes involved in the floral development exhibited remarkable increases, especially in older galls. It seems that the gene expressions noted in connection with galls would be in agreement with the notion that it is the vascular cambium that gives rise to the meristematic tissue and alters leaf development towards the formation of carpels. Gene regulation in floral reversions has been studied by Wang et al. [39], who showed that plant hormone biosynthesis and signal transduction, starch and sucrose metabolism, DNA replication and modification, as well as other processes crucial for switchgrass flower reversions, were controlled by the genes earlier identified by the researchers.

In a recent publication, Korgaonkar et al. [40] suggested that the colour of galls produced by the aphid *Hormaphis cornu* on the leaves of *Hamamelis virginiana* (and maybe other aspects of the gall’s development as well) could be caused by specific secretions of the galling aphid. The researchers found that derived genetic variants in the aphid gene determinant of gall colour “are associated with strong downregulation of dgc transcription in aphid salivary glands, upregulation in galls of seven genes involved in anthocyanin synthesis, and deposition of two red anthocyanins in galls”. A small number of plant genes may, thus, be responding. The proteins secreted by the gall insect in its saliva belong to a large and diverse family, for whose members a pair of widely-spaced cysteine-tyrosine-cysteine residues is typical [40].

## 3. The Suggestion to Solve the Problem

Just as it is with edible insects where not all species are equally useful as a source with which to augment food supplies, it is with plant galls: not all will be equally useful when it comes to promote plant galls as a source of food or feed or as a raw material to extract pharmaceuticals from. Already as a student I was fascinated by plant galls [41] and I now believe especially the larger and fleshy kinds of galls can serve as an example of what I have in mind. There are numerous so-called “oak apples”, as these galls are commonly known by. Fully developed, these spherical galls can vary in size between one and three centimetres in diameter and may differ in colour (depending on the gall-inducing species of gall wasp involved) between shades of green and red [42,43]. Although they may appear enticing and contain juicy and spongy but usually quite bitter tissue, the galls are anything but tasty as anyone who has tried to eat them will have found out. However, a few varieties of plant galls have apparently found acceptance by human consumers: some are used as drugs [44] and some, such as the galls of *Salvia pomifera* according to Ion [45], quoting the Greek botanist Eleftherios Dariotis, are appreciated by local folk, because of the galls’ pungent, herbal taste.

Generally, however, galls are not appreciated as a food item although birds such as pheasants have repeatedly been reported to consume plant galls [46,47]. Nonetheless, acceptability by humans could conceivably be changed as there are a variety of means by which improvements could be achieved. The first step would have to be to prepare a detailed analysis and understanding of the chemicals that are involved in triggering the formation of the gall by the plant. This is no easy task and such chemical inducers must be expected to be specific to particular insect species. It is therefore important to focus on some common, large, and fleshy galls for which it should be possible to identify the inducing insect species and the chemical signals that the host plant receives and responds to in order to build the gall. This task completed, the next step would then have to be the determination of the suitable dose and the duration of the inducing chemical’s presence in or on the plant that can produce a large gall. The best way to identify the genes that are involved in the gall formation is the RNA-sequencing method, which shows the expression of all genes over time during gall growth. Later it should be possible to study more aspects of those genes by gene expression analysis using real time PCR [48] and gene silencing [49] to confirm the involvement of specific genes in the process.

The final step would be to transfer the relevant genes to naïve plants, which could be possible by using plasmids or modified viruses. Gene transformation and genetic engineering has indeed contributed to increases in crop productivity [50] and methods to transfer foreign genes to plants have been summarised by, for example, by Narusaka et al. [51]. The PhD thesis by Boyer [52] touches many aspects of molecular engineering within the realms of agro-ecosystems and focuses on the transition from peasant farming via Mendelian genetics to molecular transformations without ignoring unintended consequences. For plant species grown for their biomass or fuel value a report summarizing the possibilities of engineered high energy crops has already been published by the U.S. Department of Energy [53] and a focus on plant galls should be able to generate a similar report.

## 4. The Prospect for the Future

If I have now suggested that a molecular approach can be used to grow bio-engineered ‘designer plant galls’ of different colorations, larger than normal size and desirable content, then this need not be science fiction, but something that is in our grasp. Some plant galls have been medicinally important for centuries [43,54] and are known to be rich in anti-cancer and anti-diabetic substances [55]; others are known to be rich in resins and tannic acid and have been used for dyeing, for leather treatment, as a source for ointments and for the manufacture of permanent inks [56]. Although the controlled production of plant galls is not yet possible and has to be seen as an ambitious undertaking, it is in principle achievable, even if long times are required. Once the mechanism is fully understood by which the plant is manipulated by the gall-inducer, it could then ultimately lead to an entirely new kind of agricultural activity. In addition, once that has been achieved, galls of desired sizes, coloration and content should then be able to be grown. Of course there may well be an attitude of neophobia and, at least initially, the public’s concern of a genetically modified food item, but such concerns are not new and in western countries have been dealt with in connection with sushi and edible insects [57].

Not being the result of pollination by insects, some galls could be marketed as a kind of “vegetable” or “fruit”, while others might serve the pharmaceutical industry, for example as an alternative tannin source [58]. Methanol and acetone extracts from galls of *Quercus infectoria*, used traditionally to treat wound infections after childbirth by local folk in Malaysia and to alleviate toothache and gingivitis in India, were tested for their antibacterial properties. *Streptococcus salivarius* was found to be the most susceptible and it was therefore concluded that these galls could be considered effective phytotherapeutic agents [59]. It had even been suggested as far back as 1979 that it “may be possible to adapt the transfer process [of the genetic material] in order to introduce desirable genetic traits-such as the ability to fix nitrogen-into plants.” [34].

There is a similarity to the natural process of pollination by insects and that which is simulated by manual or artificial pollination, because the natural production of plant galls depends also, at least initially, on the inputs of live insects or some other natural agent before it can be wholly controlled by humans. However, when the process of inducing a host plant to grow a gall is fully understood and can be copied and controlled, then plants can be manipulated to produce galls without the involvement of another organism. Ultimately it should be possible to grow bio-engineered plant gall material via genetic manipulation as products generated solely by tissue culture. In view of the much publicised dependence of our crops on pollinating insects on the one hand and the general decline of the insect diversity on the other, harvestable plant growths such as galls, produced without the involvement of insects, could then become a “game changer”.

## Figures and Tables

**Figure 1 foods-11-04014-f001:**
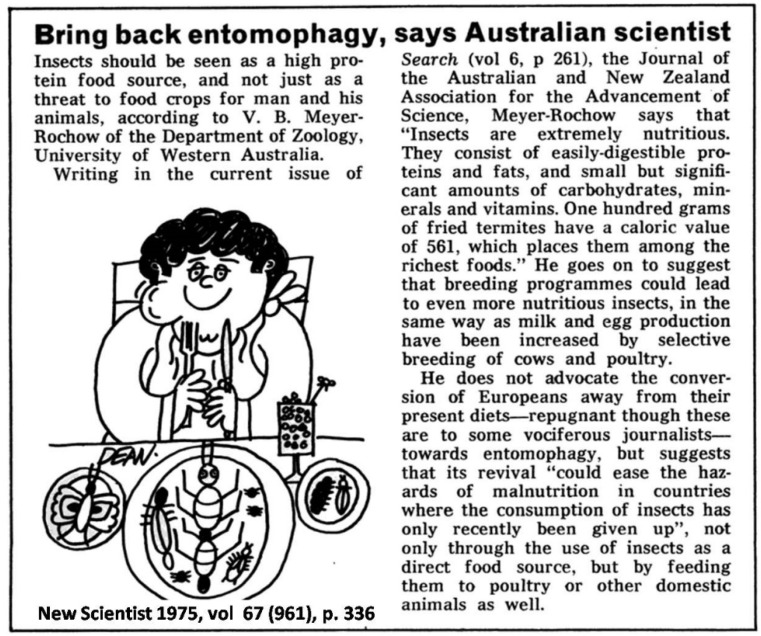
One of many cartoons that appeared in newspapers and magazines [6] after it had been suggested in 1975 that insects could ease the problem of global food shortages.

**Figure 2 foods-11-04014-f002:**
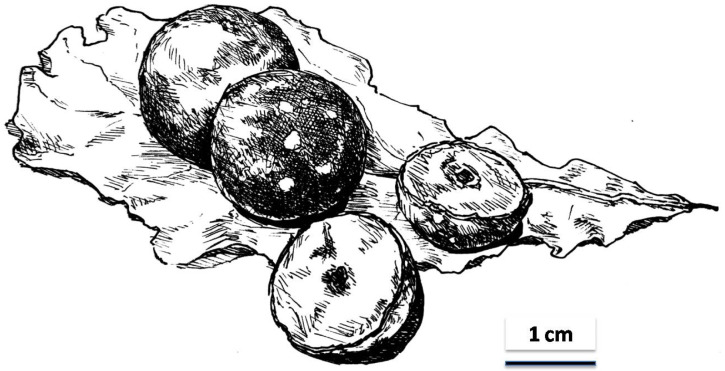
Drawing of mature and bisected oak tree galls, known as oak apples, on an oak leaf.

## Data Availability

Not applicable.

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
