# Peer review of "Can Molecularly Engineered Plant Galls Help to Ease the Problem of World Food Shortage (and Our Dependence on Pollinating Insects)?"

_foods, 2022, doi:10.3390/foods11244014_

Round 1

Reviewer 1 Report

Dear Authors,

Please find below and attached my comments and suggestions for your work.

Good luck!

Kind regards,

The Reviewer

Review Report Form

Partea superioară a machetei

Journal: Foods (ISSN 2304-8158)

Manuscript ID: foods-2006362

Type: Article

Title: Can molecularly engineered plant galls help to ease the problem of world food shortage (and our dependence on pollinating insects)?

Authors: Victor Benno Meyer-Rochow *

Section: Food Security and Sustainability

Special Issue: Role of Underutilized and Lesser-Known Foods in Achieving Nutritional Security

Submission Date: 18 October 2022

Dear Authors,

I have carefully analyzed your article entitled “Frontier Research of Management Sciences: Business Analytics, Prediction Markets and Customer Relationship Management”.

Congratulations for your work and valuable insights reflected in the content of the manuscript!

The structure of my Review Report Form takes into consideration two sections, namely: (A.) General overview of the article and strong points; and (B) Suggestions meant to improve your current manuscript.

(A.) General overview of the article and strong points:

Ø  General background of the study and aim of the paper: The authors have mentioned that the world faces numerous problems and two of them are global food shortages and the dwindling number of pollinating insects. In addition, the authors mentioned that the plant products that do not arise from pollination are plant galls, which as in the case of oak apples, can resemble fruits and be the size of a cherry. It is suggested that once research has understood how chemical signals from gall-inducing insects program a plant to produce a gall, it should be possible to mimic and to improve nature and “bioengineer” designer galls of different sizes and specific contents to serve as food or a source of medicinally useful compounds. Also, the authors stressed that there is then even the prospect of engineered plant galls to be produced by plant tissue culture via genetic manipulation without the involvement of insects’ altogether.

(B) Suggestions meant to improve your current manuscript:

Distinguished Authors I would kindly like to suggest the following aspects:

(1.) Closely analyzing the article, since there are some English language improvements and slight corrections that need to be taken care of. Thus, my recommendation would be to carefully proofread the entire manuscript. 

(2.) Also, I have closely analyzed the format of the article, in order to check whether it follows the guidelines which are specific to the publisher. Thus, I have noticed that the current form of your work needs improvement in this regard. So, my kind suggestion is to closely analyze again the guidelines belonging to the publisher, since the article should fit exactly the publisher’s guidelines. For instance, the keywords, the subsections, the references, currently do not fit the style and the requirements of the publisher. Also, it would be highly recommendable to include in the abstract of your study more highly relevant details that refer to the research objectives and the methodology used, as well as the results of the study. This would definitely be considered a plus for your scientific work.    

(3.) In continuation, the suggestion would also be inserting in your article a few ideas concerning the correlation between effects of the COVID-19 pandemic and the COVID-19 global crisis, sustainability and sustainability assessment, Sustainable Development Goals, while focusing on the how can molecularly engineered plant galls help to ease the problem of world food shortage (and our dependence on pollinating insects), since these are key focuses these days. In this context, I had the chance to read a few interesting scientific works recently, among which I would like to mention: (2019). The Social, Economic, and Environmental Impact of Ecological Beekeeping in Romania. In G. Popescu (Ed.), Agrifood Economics and Sustainable Development in Contemporary Society (pp. 75-96). IGI Global. https://doi.org/10.4018/978-1-5225-5739-5.ch004. https://www.igi-global.com/chapter/the-social-economic-and-environmental-impact-of-ecological-beekeeping-in-romania/210016; OECD. Measuring the Impacts of Business on Well-Being and Sustainability. https://www.oecd.org/statistics/Measuring-impacts-of-business-on-well-being.pdf; OECD. 2022. Toward sustainable economic development through promoting and enabling responsible business conduct. https://www.oecd-ilibrary.org/sites/f7813858-en/index.html?itemId=/content/component/f7813858-en.

I am looking forward to receive your comments and to read the new and improved version of your work.

Dear Authors, congratulations once again for your work and valuable insights reflected in the content of the manuscript, and I hope my comments will be of value to you!

Kind regards,

The Reviewer

Partea inferioară a machetei

Author Response

Reviewer 1

It was very satisfying that all four reviewers in connection with this manuscript used words like ”interesting”, “curious” and “refreshing” and supported its publication. All four suggested additional references (nine additional important references were added); two felt an illustration of some plant galls would be useful (one has now been added as Fig. 2) and two thought the author should perhaps mention possible resistance of the public to GMO foods (this has been done).

Reviewer 1  felt that the Abstract should contain some more detailed information on objectives and methodology. This has now been done and the following sentences  have been added to the Abstract:  “It is suggested that once research has understood how chemical signals from gall-inducing insects program a plant to produce a gall, it should be possible to mimic and to improve nature and “bioengineer” designer galls of different sizes, colorations and specific contents to serve as food or a source of medicinally useful compounds. To achieve this objective, the genes involved in the formation of the galls need to be identified by RNA-sequencing and confirmed by gene expression analyses and gene slicing. Ultimately the relevant genes need to be transferred to naïve plants possibly with the aid of plasmids or viruses as practiced in crop productivity increases.”

Reviewer 1 wondered if the covid pandemic should not have been mentioned. We disagree and did not mention it.

Reviewer 1 suggested  to refer to an interesting chapter that provides information on additional problems in beekeeping and these were overcome in Romania. Towards that end an additional paragraph was added: “There are, of course, other factors that can affect beekeeping and bee populations, as in the case of Romania during the transition period after 1989 to a market economy, when beekeeping experienced a severe decline in that country. For reasons explained in detail by Popescu et al. in 2019 [14], it was largely because of the long history of beekeeping in Romania and the experience in economic, social and environmental contexts that apicultural practices recovered in that country. However, even that country is not immune to the global concern of declining pollinator populations.”

References were revised according to instructions for authors.

Reviewer 2 Report

This is an entertaining, though highly speculative, review on the potential uses that could be made of the mechanisms of insect gall induction, once such mechanisms have been elucidated. There are no factual errors, though the author has overlooked multiple highly relevant earlier papers. I encourage the author to read the papers below and related work and incorporate this information into their review.

90 – Several papers show that galls are powerful nutrient sinks, and therefore suggest that insect galls exact a considerable physiological cost on plants. For example,

Inbar, Eshel, and Wool, 1995 “Interspecific Competition among Phloem-Feeding Insects Mediated by Induced Host-Plant Sinks.”

134-136 – There is no evidence I am aware of that gall inducing insects transfer DNA to the plant to induce galls. It is also difficult to imagine a mechanism by which this might occur.

137 – 139 – There are multiple better examples supporting the idea that galls are extended phenotypes of the gallers. See the following

Dodson, 1991 “Control of Gall Morphology: Tephritid Gallformers (Aciurina Spp.) on Rabbitbrush (Chrysothamnus).”

Stern, 1995 “Phylogenetic Evidence That Aphids Rather than Plants, Determine Gall Morphology.”

Crespi and Worobey, 1998 “Comparative Analysis of Gall Morphology in Australian Gall Thrips: The Evolution of Extended Phenotypes.”

176 – The author has apparently missed this publication, which provides some evidence of the potential mechanisms of gall induction by aphids.

Korgaonkar et al., 2021 “A Novel Family of Secreted Insect Proteins Linked to Plant Gall Development.”

Author Response

It was very satisfying that all four reviewers in connection with this manuscript used words like ”interesting”, “curious” and “refreshing” and supported its publication. All four suggested additional references (eight additional important references were added); two felt an illustration of some plant galls would be useful (one has now been added as Fig. 2) and two thought the author should perhaps mention possible resistance of the public to GMO foods (this has been done).

Reviewer 2 provided information on very important earlier publications which had been missed. I thank this reviewer for pointing out these very informative and essential papers, which have now all been mentioned in the appropriate places of the ms.  The papers that were suggested by the reviewer and had been missed were really important and I am really grateful to the reviewer.

This paragraph was added: “In a recent publication, Korgaonkar et al. [40] suggested that the colour of galls produced by the aphid Hormaphis cornu on the leaves of Hamamelis virginiana (and maybe other aspects of the gall’s development as well) could be caused by specific secretions of the galling aphid. The researchers found that derived genetic variants in the aphid gene determinant of gall colour “are associated with strong downregulation of dgc transcription in aphid salivary glands, upregulation in galls of seven genes involved in anthocyanin synthesis, and deposition of two red anthocyanins in galls.” A small number of plant genes may, thus, be responding. The proteins secreted by the gall insect in its saliva belong to a large and diverse family, for whose members a pair of widely-spaced cysteine-tyrosine-cysteine residues is typical [40].”

Reviewer 3 Report

Quite interesting and refreshing idea, not realistic by this moment but maybe it will spark some future studies.  Besides:

Line 44

Why don't you cite to directly Giannini et al. 2017?

Line 69

It would definitely look better If you wrote “summarised by Althaus et al [9]” than “summarised in [9]”

Line 71-74

Some references should appear here or are these only your suppositions?

Line 78-79

Some references should appear here

Author Response

It was very satisfying that all four reviewers in connection with this manuscript used words like ”interesting”, “curious” and “refreshing” and supported its publication. All four suggested additional references (eight additional important references were added); two felt an illustration of some plant galls would be useful (one has now been added as Fig. 2) and two thought the author should perhaps mention possible resistance of the public to GMO foods (this has been done).

Reviewer 3 suggested that I cite the important paper by Giannini et al. 2017 (which I now do). I have to point out to Reviewer 3 that I agree: some of the thoughts are not realistic at present and, in the words of the reviewer “will spark some future studies”. However, the ideas are not ‘science fiction’ like time travel or tele-transportation, but deal with achievable goals.

Reviewer 4 Report

The manuscript is clear and well-written, and the topic is curious and interesting. Nevetherless, I propose some changes:

- L69 considerable concerns

- L152-L156. Please, correct the text formatting

- references: please, revise references according to authors instruction

I suggest to add figures of plant galls and/or a table showing the following information: type of plant gall, its use, the bioactive molecules that it contains (if known), the proper scientific reference. This could clarify the actual potential of plant galls.

In my opinion, to suggest that plant galls could help to ease the problem of food shortage is too ambitious. They are a curiosity, that could be eventually be a source of bioactive molecules. Very few information about their edibility is reported in the text (are plant galls actual edible?). Moreover, genetic engeneering of tree species (es. Quercus) is quite difficult and long times are required, at least, to obtain a plant able to produce galls of a certain size, from which to extract a reasonable amount of metabolites.

I suggest the author to better discuss the actual potentiality of plant galls, according to the title proposed and the doubts raised here. Furthermore, public concerns about GMO consumpion should be mentioned.

Author Response

x

It was very satisfying that all four reviewers in connection with this manuscript used words like ”interesting”, “curious” and “refreshing” and supported its publication. All four suggested additional references (eight additional important references were added); two felt an illustration of some plant galls would be useful (one has now been added as Fig. 2) and two thought the author should perhaps mention possible resistance of the public to GMO foods (this has been done).

Reviewer 4 pointed out that the references had to be revised and listed according to authors instructions by the journal. All references were duly revised.

A sentence regarding GMO foods and neophobia has been added (“Of course there may well be an attitude of neophobia and, at least initially, the public’s concern of a genetically modified food item, but such concerns are not new and in western countries have been dealt with in connection with sushi and edible insects [57].”  The paper dealing with such issues is referenced (“Faccio, E.; Favino, L.,G.,N. Food neophobia or distrust of novelties? Exploring consumers’ attitudes towards GMOs insects and cultured meat. Applied Science. 20199(20), 4440; https://doi.org/10.3390/app9204440.

I agree with the reviewer: some of the thoughts are not very realistic at present and, in the words of the reviewer “will spark some future studies”. However, the ideas are not ‘science fiction’ like time travel or tele-transportation, but deal with achievable goals. The text has been revised and I think it is now clear that the task to bioengineer plant galls is complex, difficult and requires time. However, it is achievable.

Round 2

Reviewer 4 Report

In my opinion, the manuscript is significally improved. Nevertheless, I still think that plant galls could hardly help to ease the problem of food shortage, so I suggest to rephrase the title.

Author Response

The reviewer had some excellent and very helpful suggestions, which I had responded to earlier. His latest suggestion concerns only the title. However, I am reluctant to change the title, for firstly, my manuscript will be published under the caterory "Comments" (not 'Research Papers').  Secondly, the reviewers skepticism is already reflected in the use of the question mark at the end of the title. I think that says it all. Of course I could 'rephrase' the title amd write "Can molecularly engineered plant galls be of use in easing the problem of world food shortage (and our dependence on pollinating insects)?" or "...assist in..." But does that really make much of a difference? The question mark is part of the title !  The third reason for choosing the title was that the 1975 paper of mine on edible insects was "Can insects help to ease the problem of world food shortage?" That particular paper was the beginning of a worldwide interest in edible insects and by using a similar title, I wanted to link this plant gall papers to the earlier edible insect paper! I